# Mental health service readiness in Nepal: Insights from the 2021 Nepal Health Facility Survey

**Kiran Acharya[1], Devendra Raj Singh[2], Anjalina Karki[3], Michelle Cleary[4], Deependra K. Thapa[3,4,5]\***

1 New ERA, Rudramati Marga, Kathmandu, Nepal , 2 School of Human and Health Sciences, University of Huddersfield, Huddersfield, United Kingdom, 3 Nepal Public Health Research and Development Center, Kathmandu, Nepal, 4 School of Nursing, Midwifery and Social Sciences, Central Queensland University, Sydney, New South Wales, Australia, 5 School of Public Health – Bloomington, Indiana University, Bloomington, Indiana, United States of America

\* deependrakajithapa@gmail.com

## Abstract

Mental health conditions have risen across the world. However, evidence on the availability and readiness of mental health services in resource-poor countries like Nepal is scarce. This study examined the readiness of health facilities to deliver mental health services and factors associated with service readiness in Nepal. The study used data from the 2021 Nepal Health Facility Survey (n = 394 health facilities offering mental health services). Aligning with the WHO's Service Availability and Readiness Assessment framework, the mental health service readiness score was determined for staff, guidelines, and medicine domains. A weighted additive model to define both domain-specific and overall readiness was used, followed by ordinary least squares and quantile regression methods to examine the factors associated with service readiness. The results showed that one-quarter of health facilities were providing mental health services. The overall readiness score was 22.2%, with domain scores of 14% for staff and guidelines, and 30% for medicines. Ordinary least squares regression showed that health facility type (lower in private facilities and basic health centres), province (lower in Madhesh province), and user fee (higher in facilities charging a separate user fee) were significantly associated with service readiness. The quantile regression demonstrated that the effect of these variables varied with the order of the quantile groups. Service readiness for mental health services provision is sub-optimal in Nepal, with few trained health workers, an absence of guidelines and protocols, and limited access to medicines.

## Introduction

Mental health conditions are reportedly increasing and are a global health concern. They are the leading cause of the global disease burden in terms of years lived with

**Data availability statement:** The datasets used in this current study are available on the Demographic and Health Survey Program repository, which is a public, open-access repository: https://dhsprogram.com/data/available-datasets.cfm.

**Funding:** The author(s) received no specific funding for this work.

**Competing interests:** The authors have declared that no competing interests exist.

disability (YLDs), accounting for more than 10% of the global Disability Adjusted Life Years (DALYs) [1,2]. Despite efforts to address mental health conditions at the local, regional, and international levels, a significant proportion of people with mental health conditions continue to face challenges in accessing services, especially in resource-poor settings [3,4]. The rising mental health conditions place a considerable burden on healthcare systems in resource-poor settings in Low- and Middle-Income Countries (LMICs) such as Nepal, in which the health system is identified as fragile and less resilient [1,3].

Nepal's most recent national mental health survey, 2020, reported the lifetime prevalence of mental disorders among the adult population to be 10%, the prevalence of any mental disorder among adolescents to be 5.2%, and the current prevalence of any mental disorders to be 4.3% [5]. Almost 16.0% of deaths among Nepalese reproductive-age women were by suicide [6]. Recent systematic reviews in Nepal have also documented elevated rates of mental health conditions in children and adolescents [7], and older adults [8].

Previous studies in Nepal have identified high unmet needs with poor access to mental health services, attributed to a combination of factors including a shortage of qualified health staff, limited availability of mental health medications, lack of privacy during treatment, mistreatment by health services, mental health stigma, and cultural taboos [9,10]. More than two-thirds of individuals with a mental health condition had initially sought treatment from a community faith healer [11]. In addition to the above, the prevailing treatment gap for mental health conditions, combined with a fragmented mental health service throughout Nepal, demonstrates the need to build capacity within health facilities to strengthen the delivery of mental health services [11].

Moreover, the Sustainable Development Goal (SDG) target 3.4 mandates the promotion of mental health and well-being to achieve the population's overall health [12]. However, in Nepal, there is only one specialised tertiary-level hospital in the capital city, which provides mental health services, with a few other government and private hospitals providing general psychiatric services in different locations across the country [13]. The deep-rooted social stigma and discrimination towards mental conditions further pose a significant challenge, resulting in delays in seeking treatment, which impacts treatment and recovery [13].

In 2017, the Nepal government introduced the Community Mental Health Care Package to support the implementation of the National Mental Health Policy [14]. This package is intended to integrate mental health into primary health care (PHC) services. It aligns with the Mental Health Gap Action Programme, developed by the WHO [15] to scale-up cost-effective interventions for mental, neurological, and substance use disorders through the capacity building of PHC workers [16]. The National Mental Health Strategy and Action Plan 2020 [17], based on the National Health Policy 2019, outlines the integration of mental health services with PHC [18]. However, implementation is still in the initial stage across Nepal.

Improving accessibility and availability of services in PHC facilities may increase service utilization [19]. There is limited evidence concerning the readiness of health facilities to deliver mental health services across Nepal. Nepal is currently

implementing the WHO Special Initiative on Mental Health [15] by introducing a comprehensive multiyear action plan to integrate and deliver mental health services across the existing health system [10,20]. Accessible, safe, effective, timely, efficient, equitable, person-centred, and recovery-oriented mental health services are essential to achieve the goal of universal health coverage [4].

Nepal is expanding basic health services to achieve universal health coverage [21], emphasizing quality of care for better health outcomes and patient satisfaction [22]. While the quality of care lacks a single definition, the Donabedian model (inputs, processes, outputs) is influential [23,24]. Service readiness (available inputs) is an essential component for quality of care [25]. Service readiness generally refers to being prepared and capable of delivering the service. Health service readiness refers to a health facility's overall capacity to deliver general health services and is defined by the availability of essential elements required for service provision, including basic amenities, necessary equipment, trained staff, standards and protocols, and medicines and supplies [26]. Previous studies have reported that service readiness is associated with better provision of care, enhanced service quality, and improved health outcomes [25,27]. Therefore, assessing and improving facility readiness is critical for strengthening health systems and achieving universal health coverage. Despite some research on barriers to mental health service use, nationally representative studies on service availability and readiness are lacking.

This study aimed to assess the readiness of mental health services in Nepal utilizing nationally representative data from the 2021 Nepal Health Facility Survey (NHFS). We adopted the World Health Organization's (WHO) Service Availability and Readiness Assessment (SARA) framework [26] for assessing service readiness, which identifies staff and guidelines, equipment, diagnostics, and medicines as the domains of service readiness. Leveraging available data from the 2021 NHFS [28], we used this framework, including the availability of staff, guidelines, and medicines, as the domains of service readiness. We further examined the facility-level factors associated with mental health service readiness. This study's findings may help policymakers address supply-side constraints and improve access to mental health services at different levels of health facilities.

## Methods

### Ethics statement

The data used in this study were obtained from a survey approved by the ICF (USA) ethics committees and the Nepal Health Research Council. This study involved a secondary analysis of publicly available data from the 2021 NHFS. The datasets used are accessible through the Demographic and Health Survey (DHS) Program's public, open-access repository: https://dhsprogram.com/data/available-datasets.cfm. Therefore, no additional ethics approval was required.

### Data source

This analysis uses data from the 2021 NHFS. The survey assessed the availability and readiness of facilities to offer basic and essential health services, with data collected between January 2021 and September 2021 (no data collected May – July due to the COVID-19 lockdown). The 2021 NHFS employed four tools: a Facility Inventory Questionnaire, a Health Provider Questionnaire, an Exit Interview Questionnaire, and Observation Checklists for antenatal care, family planning, and child health services. This study used variables from the Facility Inventory and Health Provider Questionnaires. The Facility Inventory Questionnaire gathered data on staffing, staff training, infrastructure, equipment, medicines, supplies, and services offered. The Health Provider Questionnaire collected information on staff qualifications, professional experience, working conditions, and perceptions of the service delivery environment. Details of the survey questionnaires are available elsewhere [28].

### Sample and sampling procedure

The 2021 NHFS collected data from 1,633 health facilities using equal probability systematic sampling, with stratification based on facility type within each province. After removing duplicate samples (n = 7) and excluding the 50 facilities that

either declined to participate or were closed on the survey day, the survey assessed 1,576 health facilities. The survey collected data from public and private hospitals, primary health care centers (PHCCs), health posts (HPs), community health units (CHUs), urban health centers (UHCs), and HIV testing and counseling centers across all seven provinces of Nepal. In this study, "health facilities" encompass all these facility types. For this study, only health facilities that provide mental health services (diagnosis or treatment) were included, totalling 394 health facilities. Treatment services included either inpatient or outpatient care, or both. The survey methodology details, including the sampling procedure, are reported elsewhere [28].

## Measurement of variables

The outcome variable was mental health service readiness. We defined mental health service readiness as the willingness and preparedness of health facilities to provide mental health services, as measured by the availability and functioning of support items in the two domains: trained staff and guidelines, and medicines, consistent with the WHO SARA framework [26]. We defined trained staff as health service providers who received in-service training on mental health during the 24 months preceding the survey, involving structured sessions but not individual instruction during routine supervision. Availability of guidelines was based on the presence of any guidelines for the diagnosis and/or management of mental health problems, as observed on the day of the visit. Regarding medicines, the facility must have had at least one mental health medication available on the day of the visit, such as Amitriptyline, Fluoxetine, Carbamazepine, Phenobarbitone tablets, Sodium valproate tablets, Risperidone, Alprazolam, and Diazepam injection.

To quantify the readiness of mental health services, we constructed a readiness score using a weighted additive approach that involved allocating equal importance to individual domains while accounting for differences in the number of items within each domain. This approach, which applies equal weight across domains, is suggested to produce a balanced and interpretable composite measure [29]. Firstly, we added the items for each domain and divided these by the number of items for that domain. The result was multiplied by 100 and then divided by the total number of domains in the index to provide a score for each domain. Scores from the two mental health service readiness domains were combined using this method to obtain the overall weighted additive score, ranging from 0 to 100. A higher score indicates better readiness to provide mental health services.

The outcome variable was examined against several potential explanatory/independent variables that may affect health facilities' readiness to deliver mental health services. These variables included facility type, managing authority, ecological region, province, urbanization, quality assurance, staff management meeting, meeting with management committee members, system to gather client feedback, external supervision, and user fee. We excluded the managing authority from the regression analysis because it exhibited collinearity with facility types. These key variables were identified from the available studies which demonstrated their association with service readiness [30–32].

Facility types were categorized as public hospitals, private hospitals, PHCCs, and basic health care centres (BHCCs). BHCCs include HPs, UHCs, and CHUs. Similarly, the managing authority of the facility included public or private/NGOs. The ecological regions included Mountain, Hill, and Terai. Urbanization refers to the location of the facility and is categorized by the urbanization settings of Nepal: metropolitan/sub-metropolitan city, municipality, and rural municipality. Quality assurance was measured as the availability of a record of any quality assurance activities and categorized as performed or not performed. Performed quality assurance activities referred to as routinely conducting quality assurance activities and had documentation of recent activity. This documentation could include a report or minutes from a quality assurance meeting, a supervisory checklist, review and audit records, or quality assurance registers. Staff management meetings are routine staff management meetings, categorized as never, sometimes (less than once a month), or regularly/monthly. Management meetings with members of the facility management committee included a management meeting at least once every 6 months, with documentation of a recent meeting. Having a system to gather client feedback is categorized as absent (no) or present (yes). External Supervision in the last 4 months is categorized as no or yes. User fee refers to

whether the facility has any routine user fees for services provided to the client, which were categorized as no user fee, charge fee for separate services, and fixed fee covering all services.

## Statistical analysis

We used facility weights to restore the actual representativeness of the sampled facilities. The analysis accounted for the complex NHFS sampling design using Stata's 'svy' commands. The analysis used Stata version 17.0 [33].

The proportions of facilities with mental health service availability and individual items in the readiness domain are presented. The association between explanatory variables and service readiness score was analysed using both ordinary least squares (OLS) and quantile regression methods. We found that the readiness scores had a non-normal distribution, skewed to the right (see the boxplot in Fig 1). In such cases, quantile regression is preferred as it does not assume a normal distribution of the outcome variable and is robust to outliers and heteroskedasticity [34,35]. The heteroskedasticity test using the Breusch-Pagan test statistic also showed a significant difference from zero (p-value < 0.001), further justifying the use of quantile regression. Quantile regression models the conditional quantiles (e.g., 25th, 50th, and 75th percentiles) of the outcome variable, rather than just the mean as in OLS. This approach allowed us to examine the heterogeneous associations of covariates across different quantiles of the conditional readiness score distribution. This is particularly useful in health systems research where average effects may mask important variation across performance levels. For practical illustration, estimating the 25th percentile (lower quantile) tells us how predictors influence facilities with relatively low readiness, while the 75th percentile reflects those with high readiness.

The quantile regression allows for examining the association of a covariate on different points across the entire distribution of the readiness score (i.e., upper, lower, and median), as opposed to the OLS method, which only uses the mean of the outcome variable [36]. As a non-parametric method, quantile regression has no specific assumption concerning residual distributions. In contrast, OLS regression is sensitive to extreme values of the residuals, whereas quantile regression

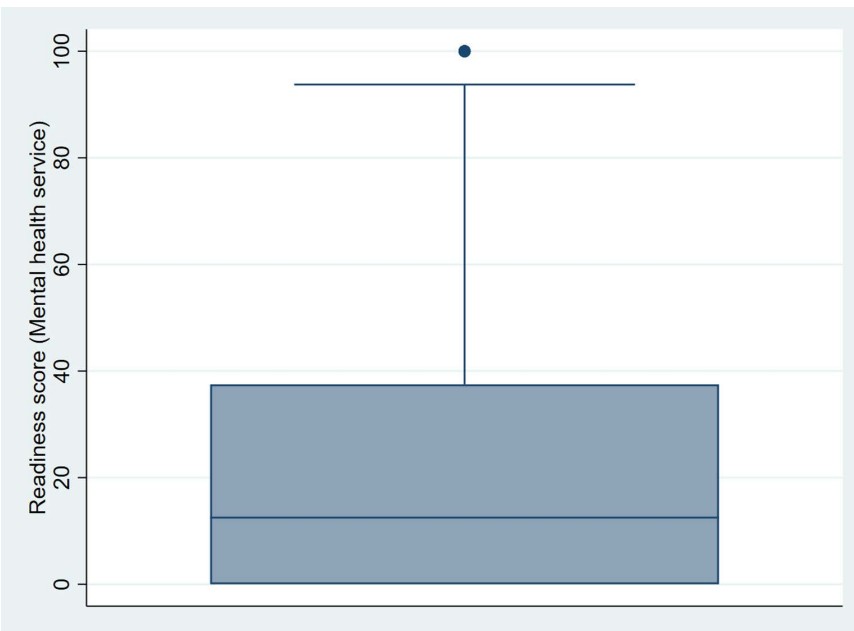

**Fig 1. Box plot (skewed right) of mental health service readiness score (%).**

is robust to extreme values of particular distributions of residuals. We performed quantile regression for the readiness score at 25%, 50%, and 75% percentiles.

## Results

### Availability of mental health services

One-quarter (25%) of health facilities were providing mental health services for either diagnosis or treatment. This is our unit of analysis for service readiness. Only 15% of facilities were found to provide both diagnosis and treatment services for mental health (Table 1).

We analyzed the distribution of mental health services (diagnosis or treatment) among facilities that provide antenatal care (ANC) services, delivery and newborn care services, non-communicable disease services (NCDs), HIV and AIDS services, and tuberculosis services (Table 2). Mental health services were available in 25% of facilities providing ANC services, 34% of facilities providing delivery services, 26% of facilities providing NCD services, 29% of facilities providing tuberculosis (TB) services, and nearly two-thirds (67%) of facilities providing HIV testing and counseling.

### General characteristics of facilities providing mental health services

Of the facilities providing mental health services, three-fifths (60%) were BHCCs, while facilities other than private hospitals constituted nearly 80%. More than half (55%) were located in hilly regions, with less than 10% in the Karnali province and the highest proportion (25%) in the Bagmati province. Around two-thirds (62%) of the facilities were in urban areas (metropolitan and sub-metropolitan cities, and municipalities). About one-third (33%) had performed quality assurance, and nearly 70% reported staff management meetings and external supervision within the last four months. Half (51%) of the facilities providing mental health services had management meetings, and 70% had a system for gathering client feedback. Around half (51%) of the facilities providing mental health services did not charge user fees (Table 3).

### Availability of supportive items for mental health services

Table 4 provides information on the availability of specific items in two domains of mental health services. These include trained staff and guidelines, and medicines available on the day of the survey. Among the facilities providing mental health

**Table 1. Distribution of mental health services among the surveyed facilities.**

| Services | n (%) | 95% CI | Total number of facilities* |
|---|---|---|---|
| Mental health services – diagnosis or treatment | 394 (25.2) | 22.3, 28.2 | 1565 |
| Mental health services – diagnosis and treatment | 227 (14.5) | 12.4, 16.8 | 1565 |

*This excludes standalone facilities providing HIV Testing and Counseling (HTC) services, as they are not intended to provide mental health services

**Table 2. Availability of mental health services among the facilities providing different specific services.**

| Services | n (%) | 95% CI | N* |
|---|---|---|---|
| Mental health services in the facilities with ANC services | 386 (25.1) | 22.2, 28.2 | 1538 |
| Mental health services in facilities with delivery services | 276 (34.4) | 30.0, 39.0 | 804 |
| Mental health services in the facilities with NCDs (CVDs or DM, or CRDs) | 394 (25.9) | 23.0, 29.1 | 1516 |
| Mental health services in facilities with HIV testing and counseling services | 50 (67.1) | 55.5, 76.7 | 74 |
| Mental health services in facilities with tuberculosis (TB) services | 369 (29.5) | 26.1, 33.2 | 1250 |

*These facilities provide specific services – ANC, delivery, NCDs, HIV, and TB.

**Table 3. Percentage distribution of facilities offering mental health services according to background characteristics (n = 394).**

| Variables | n (%) |
|---|---|
| **Facility type** | |
| Public hospitals | 41 (10.4) |
| Private hospitals | 79 (20.2) |
| PHCCs | 36 (9.0) |
| BHCCs | 228 (60.4) |
| **Managing authority** | |
| Public | 314 (79.8) |
| Private | 79 (20.2) |
| **Ecological region** | |
| Mountain | 48 (12.2) |
| Hill | 216 (54.8) |
| Terai | 130 (33.1) |
| **Province** | |
| Koshi | 53 (13.4) |
| Madhesh | 49 (12.3) |
| Bagmati | 97 (24.6) |
| Gandaki | 43 (11.0) |
| Lumbini | 72 (18.3) |
| Karnali | 35 (9.0) |
| Sudhurpaschim | 45 (11.4) |
| **Urbanization** | |
| Metro/Sub metropolitan city | 65 (16.6) |
| Municipality | 178 (45.2) |
| Rural Municipality | 150 (38.2) |
| **Quality assurance** | |
| Not performed | 265 (67.4) |
| Performed | 128 (32.6) |
| **Staff management meeting** | |
| Never/no | 31 (7.8) |
| Sometimes | 96 (24.5) |
| Regularly/monthly | 267 (67.8) |
| **Management meeting with a management committee member** | |
| No | 192 (48.8) |
| Yes | 201 (51.2) |
| **Have a system to gather client feedback** | |
| No | 119 (30.3) |
| Yes | 274 (69.7) |
| **External supervision in the last 4 months** | |
| Not occurred | 123 (31.3) |
| Occurred | 271 (68.7) |
| **User fee** | |
| No user fee | 200 (50.7) |
| Charge a fee for separate services | 174 (44.3) |
| Fixed fee covering all services | 20 (5.0) |

**Table 4. Availability of staff and guidelines, and medicines to provide mental health services (n = 394).**

| Items | n (%) | 95% CI |
|---|---|---|
| **Staff and guidelines** | | |
| Presence of guidelines | 47 (11.9) | 8.0, 17.3 |
| Availability of trained staff | 64 (16.2) | 12.1, 21.5 |
| **Medicines** | | |
| Amitriptyline | 195 (49.5) | 42.9, 56.1 |
| Fluoxetine | 93 (23.6) | 19.1, 28.8 |
| Carbamazepine | 105 (26.7) | 21.6, 32.5 |
| Phenobarbitone | 87 (22.1) | 17.8, 27.1 |
| Sodium valporate | 126 (32.1) | 26.6, 38.0 |
| Risperidone tablets | 82 (20.9) | 16.6, 26.0 |
| Alprazolam tablets | 116 (29.4) | 24.5, 34.8 |
| Diazepam injection | 149 (37.9) | 32.2, 43.8 |

services, less than one-fifth (16%) reported having at least one health provider who had received training in the 24 months preceding the survey, and 12% had guidelines related to mental health services observed on the day of the survey.

Regarding the availability of mental health medications, amitriptyline (50%) was the most readily available medication in facilities. In addition, nearly two-fifths had diazepam injection (38%), followed by sodium valproate (32%), alprazolam tablets (25%), and carbamazepine (27%). Nearly one-quarter of the health facilities had fluoxetine (24%), phenobarbitone tablets (22%), and risperidone tablets (21%) (Table 4).

### Health facility readiness to provide mental health services

Fig 2 presents the mental health readiness scores (%) for the overall facility and the specific domains. The overall readiness score was 22.2% (Mean 22.2, standard deviation: 23.2, median: 12.5). The readiness score for staff and guidelines was 14%, and medicine was 30%.

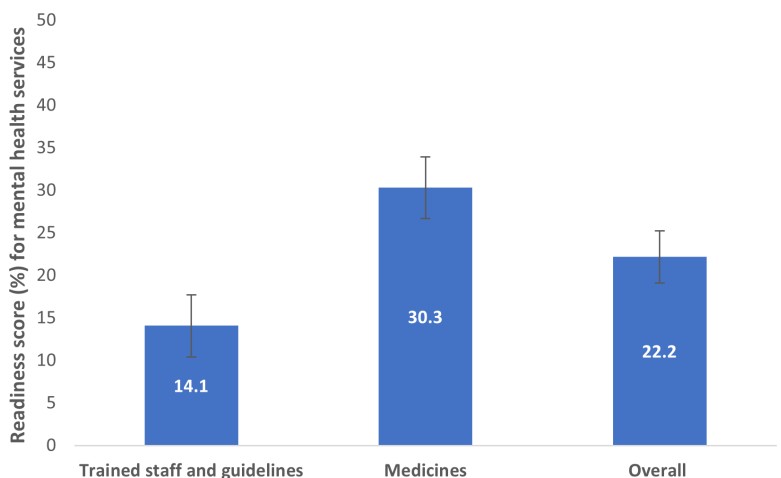

**Fig 2. Mental health service readiness score (overall, and two domains).**

**Table 5. Mental health service readiness score (%), by background characteristics (n=394).**

| Covariates | Readiness score (%) | 95% CI |
|---|---|---|
| **Total** | **22.2** | **19.1, 25.2** |
| **Facility type** | | |
| Public hospitals | 39.1 | 34.7, 43.5 |
| Private hospitals | 29.8 | 25.5, 34.1 |
| PHCCs | 31.1 | 27.4, 34.8 |
| BHCCs | 15.4 | 10.8, 19.9 |
| **Managing authority** | | |
| Public | 20.2 | 16.7, 23.8 |
| Private | 29.8 | 25.5, 34.1 |
| **Ecological region** | | |
| Mountain | 17.8 | 9.7, 26.0 |
| Hill | 24.5 | 20.2, 28.8 |
| Terai | 19.9 | 15.0, 24.7 |
| **Province** | | |
| Koshi | 16.7 | 8.9, 24.4 |
| Madhesh | 9.5 | 5.4, 13.6 |
| Bagmati | 30.2 | 22.7, 37.7 |
| Gandaki | 23.3 | 16.3, 30.4 |
| Lumbini | 21.1 | 15.7, 26.4 |
| Karnali | 16.5 | 10.0, 23.0 |
| Sudhurpaschim | 30.0 | 20.4, 39.6 |
| **Urbanization** | | |
| Metro/Sub metropolitan city | 28.1 | 21.1, 35.1 |
| Municipality | 23.7 | 19.3, 28.0 |
| Rural Municipality | 17.8 | 12.5, 23.1 |
| **Quality assurance** | | |
| Not performed | 22.5 | 18.7, 26.3 |
| Performed | 21.4 | 16.3, 26.5 |
| **Staff management meeting** | | |
| Never/no | 15.7 | 9.1, 22.3 |
| Sometimes | 19.7 | 14.2, 25.2 |
| Regularly/monthly | 23.8 | 19.8, 27.8 |
| **Management meeting with a management committee member** | | |
| No | 21.8 | 17.9, 25.6 |
| Yes | 22.5 | 17.8, 27.3 |
| **Have a system to gather client feedback** | | |
| No | 17.4 | 11.7, 23.1 |
| Yes | 24.2 | 20.6, 27.9 |
| **External supervision in the last 4 months** | | |
| Not occurred | 23.8 | 18.6, 29.1 |
| Occurred | 21.4 | 17.7, 25.1 |
| **User Fee** | | |
| No user fee | 12.6 | 8.8, 16.5 |
| Charge a fee for separate services | 33.7 | 29.5, 37.8 |
| Fixed fee covering all services | 17.0 | 9.7, 24.3 |

## Mental health service readiness score by facility-related variables

Table 5 displays the distribution of mental health service readiness scores across different facility-related background characteristics. The highest readiness score was observed in public hospitals (39%), followed by PHCCs (31%) and private hospitals (30%). The readiness score was higher in hill regions (25%), followed by terai (20%) and mountain (18%). Facilities in urban areas had a higher readiness score (28%) than those in rural areas (18%). Mental health service readiness was greater in facilities that hold regular staff management meetings, have a system for gathering client feedback, and charge separate fees for services. Surprisingly, the readiness score is higher in facilities that do not perform quality assurance or have not had supervision in the last four months, although the differences were marginal.

Fig 3 shows the provincial distribution of mental health readiness score and 95% confidence interval (CI) width. Across provinces, the highest mental health service readiness score was found in the Bagmati and Sudhurpaschim provinces (30%), while the lowest score was in the Madhesh province (9.5%) (Fig 3 and Table 5).

## Factors associated with mental health service readiness

Table 6 presents regression coefficients from OLS and quantile regression at different quantiles of the overall mental health readiness score distribution (0.25, 0.50, 0.75), assessing the association of facility-related variables with readiness score. The variables exhibited different effects based on the type of regression analysis.

In the OLS regression, the facility type, province, and user fee were associated with the mental health readiness score. Public hospitals had a significantly higher readiness score of 11.2% points (95% CI: 4.8 to 17.6) than private hospitals. Compared to Koshi province, facilities from Madhesh had a significantly lower readiness score, with a difference of -8.3% points (95% CI: -16.3 to -0.2), while facilities from Bagmati and Sudhurpaschim had significantly higher readiness scores of 12.8% points (95% CI: 3.5-22.2) and 12.3% points (95% CI: 1.1-23.5), respectively.

Facility type, province, external supervision, and user fee were significant predictors of the mental health readiness score at the 0.25 quantile. At the 0.50 (median) quantile, the mental health readiness score was significantly associated with the province and user fees. Similarly, ecological region, province, and user fee were significantly associated with the readiness score at the 0.75 quantile.

At the 0.25 quantile, the readiness score was positively associated with public hospitals and negatively associated with BHCCs. Facilities that had conducted external supervision within the last 4 months showed a significant negative effect on the readiness score. In contrast, at the 0.75 quantile, facilities located in the mountain region had a significant positive relationship with the readiness score, while user fees and province also showed significant associations with the score.

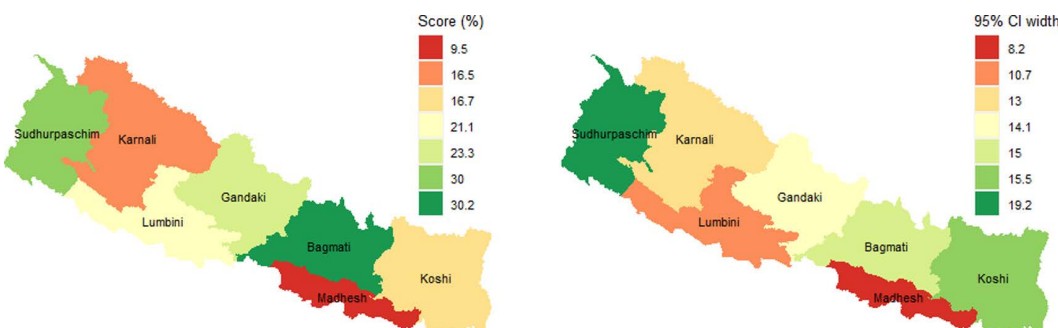

**Fig 3. Provincial distribution of mental health service readiness score and 95% CI width.** *Note: The map was generated using the R software, and the shapefile was obtained from a publicly accessible source (*https://opendatanepal.com/dataset/new-political-and-administrative-boundaries-shapefile-of-nepal*).*

**Table 6. Association (OLS and quantile regression) between different covariates and overall readiness score (n=394).**

| Variables | Unstandardized regression (adjusted) coefficients [95% CI] | | | |
| --- | --- | --- | --- | --- |
| | OLS | 0.25 quantile | 0.50 quantile | 0.75 quantile |
| **Facility type** | | | | |
| Private hospitals | ref | ref | ref | ref |
| Public hospitals | 11.2** [4.8, 17.6] | 12.5* [3.2, 21.8] | 12.1 [-1.3, 25.4] | 12.7 [-8.2, 33.6] |
| PHCCs | 4.9 [-1.9, 11.8] | 0.0 [-11.9, 11.9] | -0.9 [-17.9, 16.1] | 11.3 [-15.3, 37.9] |
| BHCCs | 2.0 [-10.0, 14.0] | -6.3* [-12.3, 1-0.3] | -8.0 [-16.7, 0.7] | 5.1 [-8.5, 18.7] |
| **Ecological region** | | | | |
| Mountain | ref | ref | ref | ref |
| Hill | 5.9 [-2.7, 14.5] | -3.1 [-7.4, 1.2] | 0.0 [-6.2, 6.2] | 14.3** [4.6, 24.0] |
| Terai | 9.5 [-2.4, 21.4] | 3.1 [-2.7, 8.9] | 4.0 [-4.3, 12.3] | 19.2 [6.3, 32.2] |
| **Province** | | | | |
| Koshi | ref | ref | ref | ref |
| Madhesh | -8.3* [-16.3, -0.2] | -5.2 [-10.8, 0.4] | -6.25 [-14.3, 1.8] | -12.5 [-25.0, 0.03] |
| Bagmati | 12.8** [3.5, 22.2] | 4.2 [-0.4, 8.8] | 7.1 [0.5, 13.7] | 22.3*** [12.0, 32.7] |
| Gandaki | 6.2 [-2.6, 15.1] | 6.3* [0.7, 11.8] | 6.3 [-1.7, 14.3] | 11.7 [-0.8, 24.2] |
| Lumbini | 4.6 [-4.0, 13.3] | 6.3* [1.4, 11.1] | 0.9 [-6.0, 7.8] | 11.4* [0.6, 22.2] |
| Karnali | 7.7 [-1.4, 16.9] | 3.1 [-3.1, 9.3] | 4.0 [-4.9, 12.9] | 11.6 [-2.4, 25.4] |
| Sudhurpaschim | 12.3* [1.1, 23.5] | 7.3* [1.7, 12.8] | 10.3* [2.3, 18.2] | 21.1** [8.7, 33.6] |
| **Urbanization** | | | | |
| Metro/Sub metropolitan city | ref | ref | ref | ref |
| Municipality | 4.9 [-2.0, 11.8] | 5.2 [-0.2, 10.7] | 3.1 [-4.7, 10.9] | 5.2 [-7.0, 17.4] |
| Rural municipality | 0.9 [-7,1, 8.9] | 2.1 [-3.5, 7.6] | 0.0 [-8.0, 8.0] | -1.4 [-13.8, 11.1] |
| **Quality assurance** | | | | |
| Not Performed | ref | ref | ref | ref |
| Performed | 0.9 [-4.8, 6.6] | 0.0 [-3.2, 3.2] | -3.1 [-7.7, 1.5] | 6.0 [-1.1, 13.1] |
| **Staff management meeting** | | | | |
| Never/no | ref | ref | ref | ref |
| Sometimes | -1.9 [-9.4, 5.6] | -1.0 [-6.6, 4.5] | -6.3 [-14.2, 1.7] | -0.8 [-13.2, 11.6] |
| Regularly/monthly | 1.2 [-5.4, 7.9] | -1.0 [-6.1, 4.1] | -5.4 [-12.7, 2.0] | 4.5 [-6.9, 16.0] |
| **Management meeting with a management committee** | | | | |
| No | ref | ref | ref | ref |
| Yes | 0.1 [-4.8, 5.0] | -2.1 [-5.2, 1.1] | 0.9 [-3.6, 5.4] | 0.6 [-6.9, 7.1] |
| **Have a system to gather client feedback** | | | | |
| No | ref | ref | ref | ref |
| Yes | 3.5 [2.6, -9.6] | 1.0 [-2.1, 4.2] | 4.5 [-0.04, 8.9] | 1.1 [-5.9, 8.1] |
| **External supervision in the last 4 months** | | | | |
| Not occurred | ref | ref | ref | ref |
| Occurred | -3.9 [-9.5, 1.7] | -3.1* [-6.0, -0.2] | -3.1 [-7.3, 1.1] | -4.5 [-11.1, 2.0] |
| **User fee** | | | | |
| No user fee | ref | ref | ref | ref |
| Charge a fee for separate services | 17.9* [7.2, 28.7] | 4.2* [0.5, 7.8] | 16.1*** [10.8, 21.3] | 23.0*** [14.8, 31.2] |
| Fixed fee covering all services | 3.1 [-5.8, 12.0] | 1.1 [-6.6, 8.7] | 1.3 [-9.6, 12.3] | 5.2 [-11.9, 22.3] |

*p<0.05,

**p<0.01,

***p<0.001 Note: CI Confidence Interval, OLS Ordinary Least Square

Based on both the conditional quantiles and mean readiness score, there was a significant and heterogeneous relationship between the province and user fees. The OLS and quantile regression coefficients across all quantiles of the readiness score showed a varying association with the province (particularly Sudhurpaschim compared to Koshi province) and readiness score, as well as between user fees and readiness score.

Fig 4 shows estimated coefficients for different explanatory variables plotted across conditional quantiles of the readiness score. The province and user fee have heterogeneous effects across the entire conditional readiness score distribution. The dotted lines represent the mean effect and coefficient (95% CI) derived from OLS regression. The green line represents the estimates, and the grey shaded area represents the CIs from the quantile regression. The CI resulting from OLS is not constant across quantiles, and quantile regression yields a coefficient (and 95% CI) that varies depending on the position in the readiness score distribution.

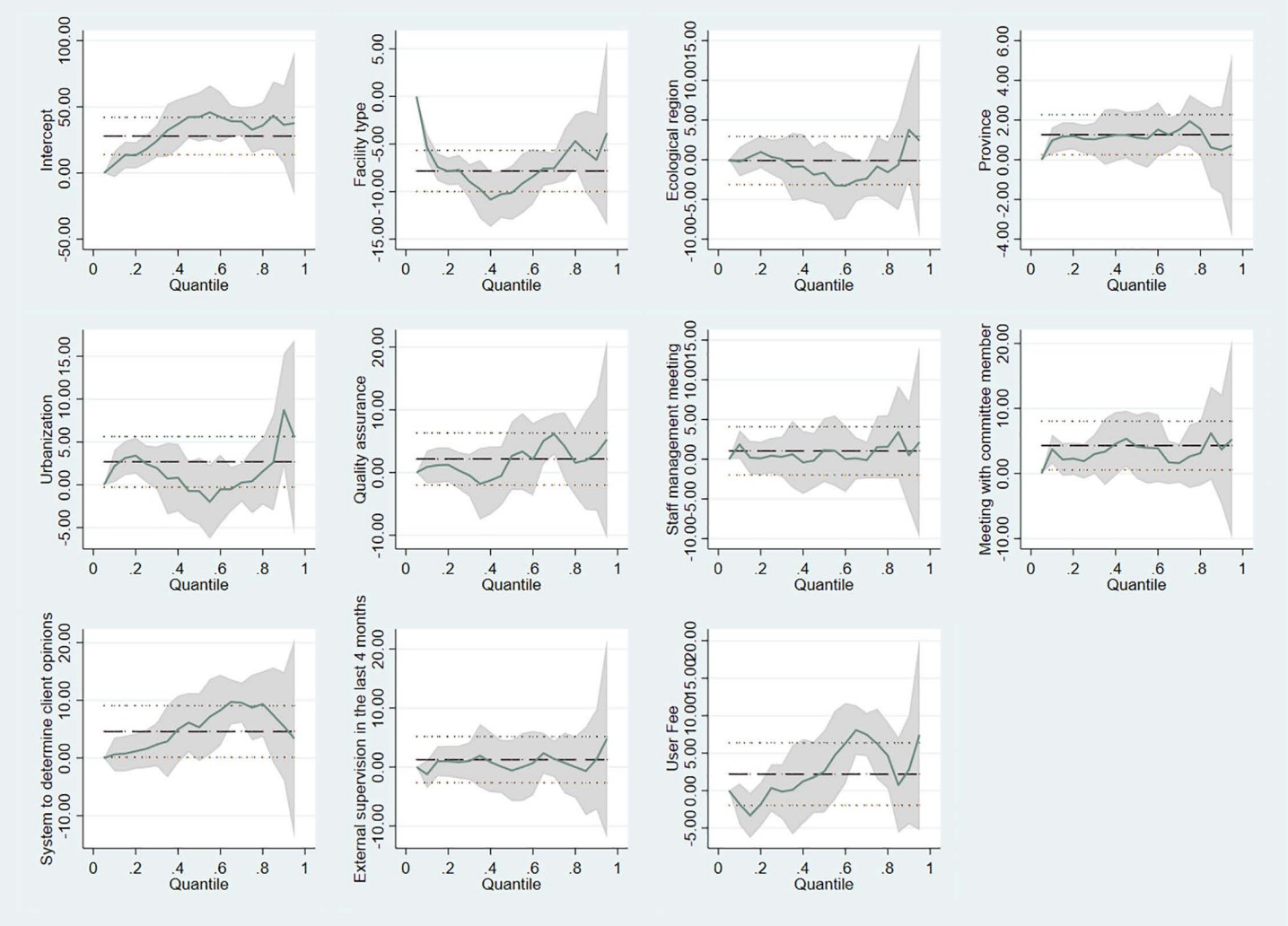

**Fig 4. Regression estimates from OLS and quantile regression across the covariates.** *Note.* The dotted horizontal lines represent the mean effect (coefficient) and 95% CI derived from OLS regression. The x-axis displays conditional quantiles of the overall mental health readiness score, representing the distribution of scores from lower (e.g., 0.2) to higher quantiles (e.g., 0.8). The y-axis shows the estimated coefficients, indicating the strength and direction of the association between a specific factor (e.g., facility type, ecological region, etc.) and the readiness score across these quantiles.

## Discussion

Despite the increasing burden of mental health, there are limited studies about the readiness of mental health services in low- and middle-income countries, including Nepal. Informed by a secondary analysis of the 2021 NHFS, this study assessed mental health services readiness, and factors associated with their readiness. The findings showed that a quarter of facilities had service availability (diagnosis or treatment) for mental health conditions, while 15% had both diagnosis and treatment. The overall readiness was 22.2%. In terms of domains of service readiness, only 16% of facilities had trained staff, 12% had guidelines in place, while less than 50% had availability of all listed medicines for treating mental health conditions.

Our findings show low overall readiness for mental health service provision, particularly in the domain of trained staff and guidelines, which mirrors broader national challenges. A recent study found that nearly half of primary healthcare providers in Nepal exhibited low mental health literacy [37], despite their critical role in community mental health service delivery. This underscores a systemic gap in the preparedness of frontline health workers and supports our finding that in-service training on mental health is limited across facilities. Additionally, the mental health workforce in Nepal remains severely under-resourced, with specialist services concentrated in urban tertiary centers. While the mhGAP initiative promotes task-sharing with non-specialist providers [16], effective rollout has been limited. The shortage of trained human resources (staff) may partly explain the low readiness scores we observed, especially in rural or peripheral health facilities. Although psychotropic medications are listed in the Basic Healthcare Package and are to be covered under the National Health Insurance Scheme, their inconsistent availability at PHC-level facilities is well-documented [38]. Our data also reflected this, with many facilities lacking essential mental health medication on the day of the assessment. These findings collectively point to an implementation gap between policy and practice.

The finding shows that only 25% of health facilities provide either diagnosis or treatment services alone, while only 15% of facilities provide both. This highlights a critical gap in Nepal's mental health service delivery. On the one hand, this reflects the limited availability of mental health services, while on the other, it underscores the lack of integrated care. This fragmentation leads to delays in treatment and discontinuation of care, ultimately resulting in poor outcomes. Facilities offering only diagnostic services should establish structured referral pathways to ensure timely access to treatment, supported by clear referral protocols, trained health workers, and follow-up mechanisms [16]. Similarly, treatment-only facilities should ensure accurate diagnosis through standardized screening tools, telepsychiatry collaboration, and task-sharing approaches where non-specialist providers are trained to diagnose and manage common mental disorders [39]. However, Nepal's referral system remains weak and unstructured, with poor coordination between primary healthcare facilities, referral hospitals, and psychiatric centers, preventing effective identification and treatment of mental illness even in facilities with trained health workers [40]. Strengthening referral pathways and integrating mental health services at all levels are crucial to providing a continuum of care and improving mental health outcomes.

Mental health services were available in less than one-third of the facilities providing ANC, NCD, TB and HIV services. While dedicated mental health facilities may exist, the reality is that many individuals access healthcare through these more general service points. Our findings highlight a crucial gap: even facilities offering these essential services often lack the basic components for mental health care, such as trained staff, guidelines, and necessary medication. This underscores the need to integrate mental health services into these healthcare pathways. Consistent with our study findings, previous research suggests that mental health services in Nepal are limited by a suitably skilled workforce, which impedes access to mental health assessment and treatment [41]. A systematic review of mental health programs in primary care in LMICs emphasized the need to invest in PHC strengthening and capacity building for health workers [42]. Other studies have also identified a lack of mental health knowledge among PHC staff, a lack of medications to treat mental illness, and inadequate infrastructure as hurdles to the access and utilization of mental health services [40,43,44].

The uneven distribution of mental health facilities in Nepal, with a higher concentration in Bagmati province and a limited presence in Karnali province, highlights significant disparities in access to care. This centralization, particularly in urban areas and the capital city, creates barriers for individuals in remote and rural regions, limiting their access to timely and appropriate mental health services. Such disparities challenge Nepal's goal of achieving universal health coverage, as equitable access to care is a fundamental principle of universal health coverage [45]. Addressing this issue requires strategic interventions, including expanding mental health services in underserved areas, strengthening community-based mental health programs, and leveraging digital health solutions such as telepsychiatry to bridge the gap between rural and urban regions.

This study further analyzed the factors associated with readiness scores using OLS and quantile regression methods. The coefficients from quantile regression were graphed against each covariate to explore the heterogeneous impacts on different points within the conditional readiness score distribution. This approach enabled the examination of how the effects of the covariates varied across the conditional quantiles of the readiness scores. The findings showed that quantile regression yields different results compared to OLS. In OLS, facility type, province, and user fee were statistically associated with mental health readiness. The quantile regression showed that the effect of these variables varied with the order of percentiles. The readiness score was relatively greater in public hospitals and PHCCs but lower in basic health care centers. This shows that mental health services are often available at higher-level government health facilities, compared to private sector facilities. The local public health facilities are more affordable and accessible, but commonly face resource limitations and lower service quality [46]. The inadequate preparedness of public health facilities in delivering mental health services can result in significant inequities, disproportionately affecting rural populations. Public facilities at the community level should strengthen infrastructure, ensure accessibility, enhance local worker capacity, and implement supportive policies [47]. Private facilities can invest in specialised training, offer affordable services, engage in community outreach, and collaborate with the public sector. In addition, sharing and exchanging health expertise between the public and private sectors and engaging the community through integrated outreach programs and co-production of the services can improve the availability and accessibility of services [48].

The readiness was associated with facility type only at the 0.25 quantile groups, indicating no difference across facility type at higher percentiles. There was a heterogeneous association between province and service readiness. The OLS results showed that Madhesh province had lower service readiness, while it was observed to be relatively higher in both the Sudurpashchim and Bagmati provinces. However, the quantile regression revealed a heterogeneous association between province and service readiness. User fees separate for mental health services had a higher readiness level than those with no user fee across all three quantile groups, with the effect being larger in the higher quantile groups. This indicates that those facilities that were ready to provide mental health services charge additional costs, limiting the affordability of the available services among low-income groups. Surprisingly, facilities reporting the occurrence of external supervision in the last 4 months had lower readiness scores compared to facilities where supervision did not occur, with the association significant at the 0.25 quantile group. This contrasts with the previous findings reporting supervision as a key contributor to improved performance of mental health services in Nepal [49]. Future studies should ascertain this finding, including variations in supervision quality, frequency, and effectiveness. In this study, the analysis also showed facility location (urban vs rural), routine quality assurance, staff management meetings, and mechanisms to address clients' feedback not associated with mental health service readiness. These aspects are important areas for future research.

## Implications

These findings have significant implications for both policy and research. The results of this study underscore the challenges faced by LMICs in delivering mental health services using Nepal as an example. There is a considerable shortfall in the readiness of health facilities in Nepal to offer adequate mental health services. This was apparent in the availability of skilled health workers, guidelines, and medicines. To strengthen health facilities to provide basic mental health services,

policymakers and health managers should ensure the availability of trained health workers, established guidelines and protocols, and necessary medicines and logistics. A critical yet often overlooked barrier is the pervasive influence of structural stigma within health and social policies, as highlighted by Gurung et al. [50], which significantly undermines mental health priorities in Nepal. This stigma is evident in discriminatory language, deviations from international legal protocols, and inadequate financing for mental health services. To overcome these challenges, policymakers must revise and harmonize policies by aligning them with international standards, integrating mental health at all policy levels, and ensuring that financial resources adequately reflect the burden of mental health conditions. The government should consider integrating mental health services at the PHC level by increasing the number of trained mid-level mental health workers, thereby improving overall mental health service readiness. Given the workforce shortage across the world, it is important to enhance the capacity of the current workforce to recognize and provide care for mental health conditions by increasing their training opportunities. Continuous professional development, training and supervision are essential to develop and support the delivery of quality and evidence-informed mental health services [51].

The frequent out-of-stock of medications for a range of mental health disorders affected confidence in health services. The availability of affordable mental health medications to treat mental health disorders remains poor, with high out-of-pocket expenditures to purchase those that are available [10]. These findings highlight the need to ensure adequate medicines are available to deliver mental health services. Facilities from Madhesh province, facilities charging user fees, and private facilities need further support to strengthen service readiness.

From a research perspective, the findings revealed that the population comprises heterogeneous sub-groups across the outcome variable, and the explanatory variables may have different impacts on the sub-groups of the population. Combining the NHFS data with demand-side data from national surveys, such as the Nepal Demographic and Health Survey, would provide a more comprehensive understanding of the factors influencing service readiness. Additionally, we recommend that future research include qualitative studies to explore the perspectives of service users and providers regarding service readiness. The study highlights the limitations of traditional regression methods (e.g., OLS), particularly when the outcome variable is not normally distributed. Different factors identified to be associated with mental health service readiness showed different effects on the quantile groups. Assuming a single rate of change (effect size) of the traditional regression methods can produce a biased result, which can be overcome by quantile regression methods [52]. An important advantage of quantile regression is that it does not assume a parametric distribution and provides an estimate of the effect of covariates across the conditional quantiles of outcome variables [53].

## Strengths and limitations

There are limited studies on the availability and readiness of mental health services in Nepal, and this study provides baseline information. The NHFS 2021 used the service readiness domains for mental health services for the first time, which was further analyzed in the current study. To our knowledge, this is the first study to assess the facility's readiness for providing mental health services in a resource-poor setting. Service readiness as the outcome variable was based on standard indicators recommended in the WHO's SARA manual. Estimations obtained were adjusted to account for the clustering effect of sampling, non-response, and disproportionate sampling. Furthermore, the association between different factors and service readiness was analyzed using the OLS and quantile regression methods, with the latter providing the association at 0.25, 0.50 and 0.80 quantile groups.

This study has some limitations. First, it only assessed mental health service readiness using three components—trained staff, guidelines, and medicines—due to data limitations in the 2021 NHFS, which did not capture other readiness components like equipment and diagnostics (e.g., screening tools). We focused on staff, guidelines, and medicines, as they are the core domains of service readiness. A facility cannot be considered truly ready to provide mental health services if these elements are not in place when a client arrives seeking care. This approach aligns with the WHO SARA framework for assessing service readiness. However, we acknowledge that this framework, and therefore our

assessment, has limitations as other factors contribute to a facility's overall capacity to deliver comprehensive care. Second, the regression models were limited by the explanatory variables available in the survey. Finally, comparing these findings to other studies is difficult due to a lack of similar research.

## Conclusion

The overall readiness score of 22.2% indicates a low level of preparedness among health facilities to provide mental health services in Nepal. This suggests that the majority of facilities lack the basic availability and ability, including trained staff, clinical guidelines, and essential medicines, required to deliver basic mental health care. For individuals seeking mental health services, this low readiness likely translates into delayed diagnosis, inadequate treatment, and poor continuity of care, exacerbating the existing burden of untreated mental illness. From a policy perspective, the readiness gap highlights a pressing need to strengthen the mental health service delivery infrastructure, particularly at the primary care level. This includes scaling up structured training for health workers, institutionalizing the use of national guidelines, and ensuring consistent availability of essential psychotropic medicines. The results of this study may inform public health management and policymakers to develop tailored interventions to improve mental health services and address the increasing burden of mental health conditions. Moreover, using a quantile regression method, this study demonstrated that the effect of various factors on mental health service readiness differed across different percentile distributions of readiness scores, expanding the application scope of quantile regression.

## Acknowledgments

The authors would like to thank the Demographic and Health Survey (DHS) Program for providing access to the dataset.

## Author contributions

**Conceptualization:** Kiran Acharya, Deependra K. Thapa.

**Data curation:** Kiran Acharya.

**Formal analysis:** Kiran Acharya, Deependra K. Thapa.

**Investigation:** Kiran Acharya, Devendra Raj Singh, Anjalina Karki, Michelle Cleary, Deependra K. Thapa.

**Methodology:** Kiran Acharya, Deependra K. Thapa.

**Project administration:** Devendra Raj Singh, Anjalina Karki, Deependra K. Thapa.

**Resources:** Devendra Raj Singh, Michelle Cleary.

**Supervision:** Michelle Cleary, Deependra K. Thapa.

**Validation:** Devendra Raj Singh, Anjalina Karki, Michelle Cleary, Deependra K. Thapa.

**Visualization:** Kiran Acharya.

**Writing – original draft:** Kiran Acharya, Devendra Raj Singh, Anjalina Karki, Deependra K. Thapa.

**Writing – review & editing:** Kiran Acharya, Devendra Raj Singh, Anjalina Karki, Michelle Cleary, Deependra K. Thapa.

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
