## [Decision Letter · Decision Letter 0]

PMEN-D-24-00362

Mental health service readiness in Nepal: Insights from the 2021 Health Facility Survey

PLOS Mental Health

Dear Dr. Thapa,

Thank you for submitting your manuscript to PLOS Mental Health. After careful consideration, we feel that it has merit but does not fully meet PLOS Mental Health’s publication criteria as it currently stands. Therefore, we invite you to submit a revised version of the manuscript that addresses the points raised during the review process.

The expert reviewer has raised a number of major concerns regarding the framework and methodology of your study, please take care to address these carefully.

We look forward to receiving your revised manuscript.

Kind regards,

Avanti Dey, PhD

Staff Editor

PLOS Mental Health

Journal Requirements:

Additional Editor Comments (if provided):

Reviewers' comments:

Reviewer's Responses to Questions

**Comments to the Author**

1. Does this manuscript meet PLOS Mental Health’s publication criteria ? Is the manuscript technically sound, and do the data support the conclusions? The manuscript must describe methodologically and ethically rigorous research with conclusions that are appropriately drawn based on the data presented.

Reviewer #1: Yes

2. Has the statistical analysis been performed appropriately and rigorously?

Reviewer #1: Yes

3. Have the authors made all data underlying the findings in their manuscript fully available (please refer to the Data Availability Statement at the start of the manuscript PDF file)?

Reviewer #1: Yes

4. Is the manuscript presented in an intelligible fashion and written in standard English?

Reviewer #1: Yes

5. Review Comments to the Author

Reviewer #1: This is an important piece of work for the Nepali health system and indeed a novel study. However, there are some areas of concern listed below:

Major comments

• The study's focus is to assess health facilities' readiness for mental health services, yet the introduction section mainly describes Nepal's mental health situation. To enhance the introduction, it would be beneficial to introduce concepts like "readiness," the framework applied, and NHFS. Providing a clear rationale or explaining the broader significance of this study within existing literature would also help readers understand the paper’s scope and purpose.

• The results indicate that 25% of facilities provided either diagnosis or treatment, with only 15% offering both. This finding is concerning and should be explored further in the discussion. For facilities that offer one of these services, what provisions should ideally be in place according to the literature? Additionally, what does the literature say about the referral mechanism in Nepal, and how robust is the system to ensure appropriate management?

• This also raises questions about the framework itself, which considers a health facility "ready" as long as any of these indicators are present. There is a risk in using these findings to suggest a high level of readiness, which may actually reflect only partial readiness. How do the authors view this issue?

• “Distribution of mental health services (diagnosis or treatment) among facilities that provide antenatal care services, delivery and newborn care services, noncommunicable disease services (NCDs), HIV and AIDS services, and tuberculosis services”. Are there health facilities in Nepal specifically designed to cater only to these services? What is the rationale for analyzing mental health service availability based on these service types? Does this imply that mental health services are provided, for instance, in antenatal or postnatal clinics? The recent framework for maternal mental health in Nepal indicates a lack of mental health services for this population, so findings like this should be interpreted with caution. I suggest that the authors provide a rationale and perhaps address this in the discussion section.

• Pg. 11: "More than half (55%) were located in hilly regions, with less than 10% in the Karnali province and the highest proportion (25%) in the Bagmati province." This finding opens a discussion on how health facilities and services in Nepal are not equitably distributed, with mental health services more centralized in cities and the capital. This poses a challenge for achieving the universal health coverage that the government aims for. It may be helpful to refer to Nepal's Public Health Act and discuss this further in the discussion section.

• "Surprisingly, facilities reporting occurrence of external supervision in the last 4 months had a lower readiness score compared to facilities where supervision did not occur, with the association significant at the 0.25 quantile group." This is indeed surprising and contrasts with the findings of Breuer et al., who identified supervision as a key contributor to improved performance in public health facilities in Nepal. I recommend that the authors explore this further in relation to the broader literature on this topic.

• “For strengthening health facilities to provide basic mental health services, policymakers and health managers should ensure sufficient availability of trained health workers, established guidelines and protocols, as well as necessary medicines and logistics.” I recommend authors to review “structural stigma” and how this impacts the overall financing, prioritizing and rolling out of services. See article by Gurung et. Al, 2023.

Minor comments

• Management and health services vary between private and public health facilities in Nepal. What does this mean for the country? What would your recommendations be for private and public health facilities?

• Page 7: “Quality assurance was measured as the availability of a record of any quality assurance activities, and categorized as performed or not performed.” What does “quality assurance activities” refer to?

• May be a need for more qualitative studies on “readiness” could be one of the recommendations.

• Pg. 20- “In terms of domains of service readiness, 16% of facilities did not have trained staff and 12% did not have guidelines in place, while less than 50% had availability of all listed medicines for treating mental health conditions.” I believe there are publications reporting challenges in mental health service delivery due to poor coordination between the health institutions (governance). This can be highlighted in the discussion section. What does this mean to health system in the federalized state? (See publications by Vaidya et. Al, 2019, Chen et. Al, 2023)

• In the methods section, the authors have mentioned using Facility Inventory Questionnaire and Health Provider Questionnaire were selected out of 4 questionnaires but have not described what these questionnaires are. A brief description on the items included in these questionnaires and its relevance could make the paper better.

• In the methods section, broad term “health facilities” has been used but what does this entail needs to be mentioned. May be their levels should be mentioned.

6. PLOS authors have the option to publish the peer review history of their article (what does this mean? ). If published, this will include your full peer review and any attached files.

**Do you want your identity to be public for this peer review?** For information about this choice, including consent withdrawal, please see our Privacy Policy .

Reviewer #1: **Yes: ** Prasansa Subba

---

## [Decision Letter · Decision Letter 1]

PMEN-D-24-00362R1

Mental health service readiness in Nepal: Insights from the 2021 Nepal Health Facility Survey

PLOS Mental Health

Dear Dr. Thapa,

Thank you for submitting your revised manuscript to PLOS Mental Health. We have now secured a report from a second reviewer as per our policy and after careful consideration of this report, we would kindly request an additional round of review and invite you to submit a revised version of the manuscript that addresses the points raised during the review process.

If you have any questions, please feel free to reach out to me. Thank you for your understanding and patience during this process. 

We look forward to receiving your revised manuscript.

Kind regards,

Karli Montague-Cardoso

Executive Editor

PLOS Mental Health

Additional Editor Comments (if provided):

Reviewers' comments:

Reviewer's Responses to Questions

**Comments to the Author**

1. If the authors have adequately addressed your comments raised in a previous round of review and you feel that this manuscript is now acceptable for publication, you may indicate that here to bypass the “Comments to the Author” section, enter your conflict of interest statement in the “Confidential to Editor” section, and submit your "Accept" recommendation.

Reviewer #2: (No Response)

2. Does this manuscript meet PLOS Mental Health’s publication criteria ? Is the manuscript technically sound, and do the data support the conclusions? The manuscript must describe methodologically and ethically rigorous research with conclusions that are appropriately drawn based on the data presented.

Reviewer #2: Yes

3. Has the statistical analysis been performed appropriately and rigorously?

Reviewer #2: I don't know

4. Have the authors made all data underlying the findings in their manuscript fully available (please refer to the Data Availability Statement at the start of the manuscript PDF file)?

Reviewer #2: Yes

5. Is the manuscript presented in an intelligible fashion and written in standard English?

Reviewer #2: Yes

6. Review Comments to the Author

Reviewer #2: General Comments:

This manuscript addresses a critical and timely issue using data from the 2021 Nepal Health Facility Survey. he study has important implications for national policy and service delivery. However, to enhance clarity and accessibility, I recommend simplifying the language to ensure broader comprehension, especially for readers from diverse backgrounds including policymakers and implementers. Please also provide clearer explanations of key terms such as the "readiness score" and the significance of "quintiles" in the context of quantile regression, ideally with practical examples or analogies

Specific Comments:

1. Ethical Considerations: The ethical considerations section currently includes references to ICF (USA) and NHRC, ensuring informed consent. Since this is a secondary analysis of publicly available data, this may be redundant. Instead, please clarify who received permission for the use of the dataset, when, and for what purpose, and from whome, particularly highlighting that no additional ethical clearance was required for this secondary analysis.

2. Data Presentation Issues:

• Table 2: Under "mental health services in facilities with HIV testing and counselling," the proportion is reported as 63.7%, yet the 95% CI is stated as 30.0–39.0%. This inconsistency should be carefully checked and corrected.

• Page 12 Sentence Clarity: The sentence “three-forth (60%) were BHCCs while 80% were public” appears incomplete. Please revise for clarity.

• Page 19, Province Comparison: You state, “Madhesh province had a significantly lower readiness score of 8.3% points,” but the 95% CI includes negative values. Please clarify whether this should read “a difference of -8.3 percentage points” to match the direction indicated by the CI.

3. Discussion – Contextualizing Findings with National Initiatives: The discussion rightly notes the lack of national studies on mental health service readiness. However, it would be strengthened by comparing the readiness scores with relevant literatures addressing different components of existing national programs and packages, such as:

a. Mental health training levels and competencies among healthcare workers OR mental health literacy among healthcare providers

b. Availability and distribution of psychotropic medications under the Basic Healthcare Package and National Health Insurance

c. Human resources for mental health (psychiatrists, psychologists, and mhGAP-trained providers)

Including such comparisons could help frame your findings more meaningfully within the existing policy landscape.

4. Conclusion – Interpretation of the 22.2% Readiness Score: The conclusion repeats the main results rather than reflecting on their implications. What does a 22.2% readiness score mean for service delivery or the likelihood of adequate care for those seeking mental health services? A more interpretive and policy-relevant framing would greatly benefit the readers.

Additional Suggestions for Improvement:

• Title & Abstract: Consider briefly defining or contextualizing “readiness” in the abstract to help orient readers unfamiliar with the term.

• Figures and Tables:

• Ensure all figures (e.g., Figure 2 and Figure 4) have clear, self-explanatory legends. The regression graphs could also benefit from a brief textual description of what the axes and lines represent, especially for non-statistical audiences.

• In reference to Figure 3 a supplementary table or figure showing readiness components by province or facility type could offer more actionable insight for sub-national planning.

7. PLOS authors have the option to publish the peer review history of their article (what does this mean? ). If published, this will include your full peer review and any attached files.

**Do you want your identity to be public for this peer review?** For information about this choice, including consent withdrawal, please see our Privacy Policy .

Reviewer #2: No

---

## [Editor Report · Decision Letter 2]

Mental health service readiness in Nepal: Insights from the 2021 Nepal Health Facility Survey

PMEN-D-24-00362R2

Dear Dr Thapa,

We are pleased to inform you that your manuscript 'Mental health service readiness in Nepal: Insights from the 2021 Nepal Health Facility Survey' has been provisionally accepted for publication in PLOS Mental Health.

Best regards,

Zahra Al-Khateeb, Ph.D

Staff Editor

PLOS Mental Health